# Suppression of Ventilation-Induced Diaphragm Fibrosis through the Phosphoinositide 3-Kinase-γ in a Murine Bleomycin-Induced Acute Lung Injury Model

**DOI:** 10.3390/ijms25126370

**Published:** 2024-06-08

**Authors:** Li-Fu Li, Chung-Chieh Yu, Chih-Yu Huang, Huang-Pin Wu, Chien-Ming Chu, Ping-Chi Liu, Yung-Yang Liu

**Affiliations:** 1Division of Pulmonary and Critical Care Medicine, Department of Internal Medicine, Chang Gung Memorial Hospital, Keelung 20401, Taiwan; lfp3434@cgmh.org.tw (L.-F.L.); ycc@cgmh.org.tw (C.-C.Y.); hcu121@yahoo.com.tw (C.-Y.H.); whanpyng@cgmh.org.tw (H.-P.W.); rocephen@cgmh.org.tw (C.-M.C.); ewind14@cgmh.org.tw (P.-C.L.); 2Department of Internal Medicine, Chang Gung University, Taoyuan 33302, Taiwan; 3Department of Respiratory Therapy, Chang Gung Memorial Hospital, Keelung 20401, Taiwan; 4Community Medicine Research Center, Chang Gung Memorial Hospital, Keelung 20401, Taiwan; 5Chest Department, Taipei Veterans General Hospital, Taipei 112201, Taiwan; 6School of Medicine, Faculty of Medicine, National Yang Ming Chiao Tung University, Taipei 112304, Taiwan; 7Institute of Clinical Medicine, School of Medicine, National Yang Ming Chiao Tung University, Taipei 112304, Taiwan

**Keywords:** apoptosis, diaphragm fibrosis, phosphoinositide 3-kinase-γ, ventilator-induced diaphragm dysfunction

## Abstract

Mechanical ventilation (MV), used in patients with acute lung injury (ALI), induces diaphragmatic myofiber atrophy and contractile inactivity, termed ventilator-induced diaphragm dysfunction. Phosphoinositide 3-kinase-γ (PI3K-γ) is crucial in modulating fibrogenesis during the reparative phase of ALI; however, the mechanisms regulating the interactions among MV, myofiber fibrosis, and PI3K-γ remain unclear. We hypothesized that MV with or without bleomycin treatment would increase diaphragm muscle fibrosis through the PI3K-γ pathway. Five days after receiving a single bolus of 0.075 units of bleomycin intratracheally, C57BL/6 mice were exposed to 6 or 10 mL/kg of MV for 8 h after receiving 5 mg/kg of AS605240 intraperitoneally. In wild-type mice, bleomycin exposure followed by MV 10 mL/kg prompted significant increases in disruptions of diaphragmatic myofibrillar organization, transforming growth factor-β1, oxidative loads, Masson’s trichrome staining, extracellular collagen levels, positive staining of α-smooth muscle actin, PI3K-γ expression, and myonuclear apoptosis (*p* < 0.05). Decreased diaphragm contractility and peroxisome proliferator-activated receptor-γ coactivator-1α levels were also observed (*p* < 0.05). MV-augmented bleomycin-induced diaphragm fibrosis and myonuclear apoptosis were attenuated in PI3K-γ-deficient mice and through AS605240-induced inhibition of PI3K-γ activity (*p* < 0.05). MV-augmented diaphragm fibrosis after bleomycin-induced ALI is partially mediated by PI3K-γ. Therapy targeting PI3K-γ may ameliorate MV-associated diaphragm fibrosis.

## 1. Introduction

Acute respiratory distress syndrome is a severe catastrophic illness with a high mortality rate of 27–45% [1,2]. Patients in intensive care units (ICUs) with this syndrome often require mechanical ventilation (MV) for oxygenation. Ventilator-induced lung injury is delineated by an uneven lung inflammatory response after exposure to mechanical stretching; this is followed by a fibroproliferative reaction with extracellular matrix (ECM) deposition, ultimately leading to ventilator-induced lung injury-associated lung fibrosis [1,2,3,4]. High-intensity mechanical stretch and a local upregulation of transforming growth factor-β1 (TGF-β1) have been demonstrated to induce fibroblast to myofibroblast transdifferentiation [1,2]. The injurious effects of MV on the ECM are triggered by altered transpulmonary pressure, uneven distribution of ventilation, and enhanced lung stretching and are exacerbated by diminished pulmonary lymphatic drainage [1,4,5]. In addition to causing ventilator-induced lung injury, MV can result in diaphragmatic weakness and the rapid decline of diaphragm muscle contractility and endurance, a phenomenon commonly referred to as ventilator-induced diaphragm dysfunction (VIDD) [6,7,8,9]. Diaphragm muscle dysfunction can aggravate underlying respiratory insufficiency, contributing to the disability burden of critically ill patients. Myofiber loss and replacement of fibrotic tissue due to aberrant repair are characterized by excessive accumulation of ECM components, such as collagen; the accompanying increase in muscle fragility and stiffness can prevent the diaphragm from achieving the optimal excursion lengths required for normal respiration [10,11]. The factors that determine the development of diaphragm injury, regeneration, and progressive fibrosis leading to diaphragm weakness are unclear.

Diaphragm fibrosis is the result of abnormal accumulation of ECM components, including collagens, either due to increased ECM generation, reduced ECM degradation activity, or a combination of both [12,13]. Excessive ECM deposition can be observed in the endomysium and perimysium of the skeletal muscles. Several growth factors, such as TGF-β1 [12,14], connective tissue growth factor, myostatin, Wnt signaling, vascular endothelial growth factor, fibroblast growth factor, epidermal growth factor, and platelet derived growth factor, have been demonstrated to enhance the progression of skeletal muscle fibrosis. As a primary profibrogenic cytokine, TGF-β1 is a potent profibrogenic factor and crucial in the control of ECM production, remodeling, and degradation [10,12,15].

MV-induced oxidative stress, inflammation, and fibrogenesis are capable of reducing diaphragm contractility and are primary causative factors in the activation of proteolytic pathways, such as muscle-specific ubiquitin E3 ligases F-box protein atrogin-1 and muscle RING-finger proteins-1 (MuRF-1) [4,9]. Research has revealed that patients with long-term MV dependency are highly susceptible to ventilator-associated pneumonia and lung fibrogenesis, which can lead to restrictive ventilatory impairment [4,6,7,16]. Therefore, research should clarify the potential molecular mechanisms responsible for acute lung injury (ALI) and diaphragm dysfunction, which hamper the weaning of patients from long-term MV.

Phosphoinositide 3-kinase (PI3K), a family of cellular lipid kinases, phosphorylates the 3-hydroxyl of the phosphatidylinositol ring to produce lipid second messengers such as phosphatidylinositol (3,4,5) triphosphate [17,18,19,20]. Several studies have researched class I PI3Ks, encompassing class IA (p110α, -β, and -δ) and class IB (p110γ) isoforms [18,19]. Binding of the complexes of class IA kinases with regulatory p85-related subunits containing SH2 is substantially enhanced through receptor tyrosine kinases; by contrast, class IB p110γ associates with G-protein-coupled receptors through its regulatory subunit p101 or p84 and G-protein subunits upon receptor stimulation [18,19]. PI3K-γ is expressed in many cell types involved in airway inflammation and fibrogenesis, including fibroblasts and endothelial cells, and may recruit the influx of leukocytes into the diaphragm [17,18,19,20]. The absence of PI3K-γ can lead to reduced leukocyte activation and recruitment, decreased angiogenesis, and diminished expression of fibrogenic markers in fibrotic lung tissue [17,18,19,20]. However, the pathogenic mechanisms, including that related to the PI3K-γ pathway, in MV-induced diaphragm fibrosis remain to be clarified.

In the current study, we employed a bleomycin-induced VIDD mouse model [1,2] to (1) evaluate the involvement of PI3K-γ expression in diaphragm fibrogenesis during MV, (2) compare oxidative stress and inflammatory cytokine generation between MV and bleomycin-induced diaphragm damage, (3) examine the involvement of PI3K-γ in VIDD through PI3K-γ homozygous knockout and inhibition with AS605240 [21], and (4) investigate the involvement of the PI3K-γ signaling pathway in bleomycin-induced diaphragm myonuclear apoptosis. We hypothesized that MV with or without bleomycin pretreatment would enhance diaphragm injury, the production of oxidants and free radicals, and diaphragm fibrogenesis through PI3K-γ pathway upregulation.

## 2. Results

### 2.1. AS605240 Mitigated the Effects of MV on Bleomycin-Induced VIDD

Mice were administered either 6 mL/kg MV (V_T_6 group) or 10 mL/kg MV (V_T_10 group) with room air for 8 h to induce VIDD. We analyzed the detrimental effects of overdistension and the consequences of treatment with intraperitoneally delivered AS605240. The physiological parameters at the beginning and end of MV are summarized in Appendix A. We monitored the mean arterial pressure of the mice and maintained normovolemic states. Transmission electron microscopy was used to explore MV- and bleomycin-induced changes in diaphragm ultrastructure. Studies have demonstrated that diaphragm myofiber atrophy due to disuse could be a crucial contributor to weaning difficulties [6,7]. Relative to the V_T_6 group and the nonventilated controls, the V_T_10 mice exhibited considerable derangement in their diaphragmatic myofibrillar structures, with enlarged lipid droplets, indistinct A-bands and I-bands, destructed Z-bands, and mitochondrial swelling (Figure 1A–E); a substantial decrement in muscle fiber diameter (Figure 1G) was also observed. The administration of AS605240 significantly reduced damage to the diaphragmatic fibers (Figure 1F,G). Ultrasonography is a noninvasive method for assessing the diaphragm in patients receiving MV [6,8]; therefore, to evaluate the effects of MV and bleomycin on diaphragm contractile function, we assessed diaphragm dysfunction in the mice by utilizing small-animal ultrasound (VEVO 2100, Visual Sonics, Toronto, ON, Canada). Relative to the V_T_6 group and the nonventilated control mice, the V_T_10 group exhibited declined diaphragm excursion and thickening fraction (Figure 1H,I). Thus, MV-mediated diaphragmatic weakness was substantially inhibited by AS605240 administration (Figure 1H,I).

### 2.2. AS605240 Inhibited Bleomycin-Stimulated, MV-Induced Diaphragmatic Inflammatory Cytokine Generation; Oxidative Stress; and Expression of Caspase-3, Calpain, Atrogin-1, MuRF-1, and PGC-1α

Research has highlighted the key role of inflammatory cytokine expression and oxygen radicals in MV-induced diaphragm damage [7,9,13]. In the current study, we observed increased levels of TGF-β1 and malondialdehyde (a marker of lipid peroxidation) in the V_T_10 group relative to those in the V_T_6 group and the nonventilated controls (Figure 2A,B). Furthermore, we performed Western blot analyses to evaluate the effects of MV on bleomycin-stimulated oxidative loads (casase-3 and calpain), the ubiquitin–proteasome system (atrogin-1 and MuRF-1), and PGC-1α (a regulator of muscle oxidative capacity) associated with VIDD (Figure 2C–G). We observed increased levels of caspase-3, calpain, atrogin-1, and MuRF-1 but a decreased level of PGC-1α in the V_T_10 group relative to those in the V_T_6 group and the nonventilated controls (Figure 2C–G). However, AS605240 administration normalized the levels of the markers.

### 2.3. AS605240 Mitigated the Effects of MV on Bleomycin-Induced Collagen Fiber Generation and Fibrogenic Markers

Masson’s trichrome staining was performed to determine the effects of MV on ECM collagen fiber accumulation. We observed more collagen fibers in the endomysium and perimysium in the V_T_10 group than in the V_T_6 group and the nonventilated control mice (Figure 3A). To identify the cell types involved in the MV-induced diaphragm fibrogenesis, we measured the expression of α-SMA, a myofibroblast marker, by using immunofluorescent staining (Figure 3B). In addition, to evaluate the effects of MV on ECM deposition, we quantified the number of collagen fibers in the ECM by considering the total diaphragm collagen content and calculated a fibrosis score by using Masson’s trichrome staining (Figure 3C,D). Our results indicate that treatment with the PI3K-γ inhibitor AS605240 substantially reduced the MV-induced increase in collagen fibers and upregulation of α-SMA (Figure 3).

### 2.4. AS605240 Mitigated the Effects of MV on Bleomycin-Stimulated PI3K-γ Protein Expression

PI3K-γ activation has been reported to play a pivotal role in regulating inflammatory cytokines and MV-augmented lung fibrosis; we thus evaluated PI3K-γ protein expression to verify the involvement of the PI3K-γ signaling pathway in VIDD and diaphragm fibrosis (Figure 4) [17,18,21]. We used Western blot analyses to measure the effects of V_T_ = 10 mL/kg on bleomycin-associated PI3K-γ protein expression in the diaphragm (Figure 4A). Relative to the V_T_6 group and the nonventilated control mice, the V_T_10 group had higher PI3K-γ protein expression (Figure 4A). This increase in PI3K-γ protein expression was substantially attenuated following AS605240 administration (Figure 4A). We subsequently performed immunohistochemistry to determine the cell types that participated in the MV-induced fibrogenesis and evaluate the effects of AS605240 on PI3K-γ activation in VIDD (Figure 4B,C). Consistent with our Western blot results, we observed a significant increase in the number of diaphragm muscle fibers positively stained for PI3K-γ in the V_T_10 group relative to those in the V_T_6 group and the nonventilated control mice. Thus, AS605240 administration prevented the V_T_10-induced expression of PI3K-γ (Figure 4B,C).

### 2.5. PI3K-γ-Deficient Mice Exhibited Reduced Bleomycin-Stimulated, MV-Induced VIDD and Diaphragm Fibrosis

We employed PI3K-γ-deficient mice to determine the involvement of PI3K-γ in MV-induced diaphragm injury; we investigated whether the recovery from diaphragm injuries and fibrogenesis following AS605240 administration was prompted by PI3K-γ expression (Figure 5 and Figure 6). The detrimental effects of MV on changes in TGF-β1 production, oxidative stress, PI3K-γ mRNA expression, diaphragm contractile function, generation of collagen fibers, fibrogenic markers, and fibrosis area in mice treated with bleomycin and MV were substantially ameliorated in the PI3K-γ-deficient mice (*p* < 0.05; Figure 5 and Figure 6).

### 2.6. Suppression of the Effects of MV on Bleomycin-Stimulated Expression of BNIP-3 and Muscle Fiber Apoptosis in PI3K-γ-Deficient Mice

BCL2/adenovirus E1B 19 kDa protein-interacting protein 3 (BNIP-3) overexpression may contribute to the worsening of cell death through apoptosis [22]. We thus measured BNIP-3 expression levels and performed terminal deoxynucleotidyl transferase-mediated dUTP-biotin nick end labeling (TUNEL) staining to evaluate the involvement of the BNIP-3 pathway in the apoptosis of diaphragm muscle fibers in bleomycin-aggravated VIDD. We observed a significant increase in the BNIP-3 expression levels and more TUNEL-positive apoptotic nuclei in the diaphragm muscle fibers in the V_T_10 group relative to those in the V_T_6 group and the nonventilated control mice (Figure 7). Notably, AS605240 administration resulted in a decrement in MV-augmented and bleomycin-enhanced BNIP-3 activities and apoptosis in the diaphragm muscle fibers. The levels of BNIP-3 activity and myonuclear apoptosis were also attenuated in the PI3K-γ-deficient mice. However, following AS605240 administration, the levels of apoptotic markers in the injured mice and PI3K-γ-deficient mice remained elevated compared to those in the control mice, indicating that other mechanistic pathways participate in apoptosis; this finding warrants further investigation. Our results indicate that PI3K-γ pathway inhibition leads to a suppression of MV-augmented and bleomycin-induced inflammatory and fibroproliferative processes in the diaphragm (Figure 8).

## 3. Discussion

In this study, we investigated the influence of MV on the diaphragm in a murine model of bleomycin-induced ALI [23]. Our results indicate that (1) bleomycin increases MV-induced diaphragmatic ultrastructural injury, reduces the cross-sectional area of diaphragm myofibers, and reduces both the diaphragm excursion and thickening fraction; (2) bleomycin augments MV-induced oxidative loads (malondialdehyde, calpain, and caspase-3) and upregulates the production of the profibrogenic factor TGF-β1; (3) bleomycin stimulates MV-induced expression of the apoptotic markers caspase-3 and BNIP-3; (4) bleomycin aggravates diaphragm muscle proteolysis through increases in atrogin-1 and MuRF-1; (5) bleomycin downregulates the mitochondrial biogenesis activator PGC-1α and upregulates the mitophagic marker BNIP-3; (6) bleomycin increases the MV-induced diaphragmatic mesenchymal marker α-SMA, total collagen production, and the diaphragm fibrosis area; and (7) bleomycin stimulates MV-induced expression of PI3K-γ in the diaphragm. Notably, these deleterious effects associated with diaphragmatic fibrogenesis were attenuated by the PI3K-γ inhibitor AS605240 or gene knockout of PI3K-γ. Therapeutic targeting of PI3K-γ may thus result in a morphological and functional improvement in diaphragm fibrosis.

VIDD was long thought to be a consequence of atrophy from diaphragm muscle disuse. However, research employing ultrasound has revealed both increased and decreased diaphragm thickness in patients with prolonged MV [24]. In animal studies, MV dysynchrony and MV application with positive end-expiratory pressure have been highlighted as factors contributing to elevated levels of connective tissue components and fibrotic change in the diaphragm [11,25]. Research has also reported an increased accumulation of ECM, particularly collagen, in the injured diaphragms of critically ill patients with MV and that diaphragm fibrosis rapidly occurs following admission to ICUs [24,26]. These results indicate that the increased stiffness and fragility of the diaphragm due to fibrosis as well as myofiber atrophy are key contributors to impaired diaphragm strength and function.

Damage to the diaphragm can progress to muscle remodeling and fibrosis due to muscular dystrophies or clinical events such as VIDD-induced myotrauma [12,27]. The ECM is an indispensable constituent of the skeletal muscles, comprising 5–10% of skeletal muscle weight and aiding in force transmission, protection, and repair of injured muscle fibers [12,15,27]. Notably, laboratory and clinical studies have observed reductions in normal diaphragm muscle accompanied by an increased proportion of abnormal muscle and connective tissue in the injured diaphragms of critically ill patients receiving MV; an increase in the ECM that partially replaces the void caused by muscle fiber atrophy is referred to as replacement fibrosis [11,24,25,26]. Diaphragm fibrosis caused by ALI and MV undermines the structural integrity and functional performance of the diaphragm; previous research has underestimated these effects as well as their detrimental effects when combined with muscle disuse atrophy [24,26]. ECM accumulation is governed by several growth factors, including TGF-β1, which plays a central role in the pathogenesis of diaphragm fibrogenesis [15].

TGF-β1 can be released from infiltrating inflammatory cells (activated M2 macrophages) or injured muscle fibers as skeletal muscle injury progresses to fibrosis [12,27]. TGF-β1 stimulates resident fibroblasts and fibro-adipogenic progenitors (FAPs) to transdifferentiate into myofibroblasts, which express α-SMA and synthesize excessive ECM proteins and lead to fibrosis [12]. Wang et al. reported that FAPs are the predominant cells that produce ECM and express ECM regulatory genes, and excessive accumulation of FAPs primarily contributes to progressive diaphragm fibrosis in a mouse model of muscular dystrophy [28]. Moreover, TGF-β1 can repress FAP apoptosis and facilitate FAP proliferation and fibrogenesis [28]. TGF-β1 can also activate the production of tissue inhibitor metalloproteinases and plasminogen activator inhibitor-1, which simultaneously suppress the ECM degradation induced by matrix metalloproteinase-2 and metalloproteinase-9 [12]. In addition, oxidative stress, inflammation, and cellular senescence play a role in the activation and proliferation of fibroblasts in the TGF-β signaling pathway, which leads to the production of ECM during the progression of fibrogenesis [29]. In our study, we observed that MV stimulated the key fibrogenic drivers TGF-β1 and malondialdehyde, which may result in diaphragm injury progressing to fibrosis (Figure 2). We also observed the classic characteristics of skeletal muscle fibrosis, including myofibrillar disorganization, muscle fiber atrophy, decreased diaphragm muscle motility and performance, increased fibrosis staining and myofibroblast expression of α-SMA, and elevated collagen content and an expanded fibrosis area (Figure 3).

We also observed increased diaphragm oxidative stress and muscle proteolysis, as evidenced by upregulation of calpain, caspase-3, atrogin-1, and MuRF-1, and increased diaphragm apoptotic cell death, as evidenced by elevated levels of BNIP-3 and caspase-3, apoptosis-positive cells, and TUNEL staining. Calpain, a calcium-dependent cysteine protease, and caspase-3 can destruct myofilament structures by cleaving myosin and actin; they also participate in the ubiquitin–proteasome system for the proteolysis of disrupted proteins [20]. BNIP-3 and caspase-3 are also involved in triggering apoptosis in muscles such as the diaphragm [24,30]. The resulting protein degradation and myocellular apoptosis ultimately lead to a disruption of the myofilament structure, reduction in muscle mass, and functional loss related to force generation. In our previous study, we demonstrated that ventilation induced increased lipid peroxidation, muscle-specific ubiquitin ligases, and apoptotic markers in the diaphragms of endotoxemic mice, which can be inhibited by enoxaparin through hypoxia-inducible factor 1α (HIF-1α) [31]. It is known that PI3K is an upstream regulator of HIF-1α activity, which transcriptionally upregulates several proteins (e.g., BNIP-3) involved in initiating apoptosis [19,31]. Thus, these similarities to the findings of the present study may imply crosstalk between pathways involving HIF-1α and PI3K. Notably, mitochondrial dysfunction is instrumental in the mechanisms underlying ICU-acquired muscle weakness, including VIDD [20]. In our VIDD murine model with bleomycin-induced ALI, we also observed that downregulation of PGC-1α and upregulation of BNIP-3 contribute to impaired mitochondrial biogenesis and aggravate mitochondrial fragmentation and mitophagy during diaphragm fibrogenesis accompanied by muscle atrophy. We have previously demonstrated that MV induced TGF-β1 generation as well as type I and III procollagen, lumican, and α-SMA gene expression in the diaphragm [32]. Other studies have reported increased oxidative stress and impaired energetic metabolism in the diaphragm in bleomycin-induced pulmonary inflammation [33] and lower glycogen content in the diaphragm in bleomycin-induced lung fibrosis [34]. However, our study is the first to investigate the mechanism underlying diaphragm fibrogenesis; we observed that bleomycin augments MV-induced diaphragmatic injury and contributes to diaphragm fibrosis in an animal model simulating critically ill patients receiving MV following ALI.

Previous animal studies have revealed that PI3K-γ activation induced epithelial–mesenchymal transition and contributed to bleomycin-induced pulmonary fibrosis [35,36,37]. However, a PI3K-γ inhibitor can attenuate bleomycin-induced pulmonary fibrosis [38]. Our recent study demonstrated that the PI3K-γ pathway mediates MV-induced aggravation of epithelial–mesenchymal transition and lung fibrosis after bleomycin-induced ALI [23]. Notably, elevated levels of PI3K-γ have been observed in the fibroblasts and lung tissues of patients with idiopathic pulmonary fibrosis [18,39]. However, the relationship between PI3K-γ and diaphragm fibrogenesis has yet to be investigated. Studies have reported that a PI3K-γ inhibitor can suppress cardiac fibrosis, a devastating form of skeletal muscle fibrosis similar to diaphragm fibrosis [40], and that tumor necrosis factor (TNF)-induced myocardial remodeling and fibrosis are mediated by the activation of PI3K-γ [41]. In the present study, we observed elevated PI3K-γ expression levels in the diaphragm tissues of mice that received MV after bleomycin-induced ALI. Administration of the PI3K-γ inhibitor AS605240 or PI3K-γ homozygous gene knockout mitigated the elevated oxidative stress and TGF-β1 activation; the heightened apoptosis, muscle proteolysis, and mitochondrial dysfunction; the increased myofibroblast differentiation, collagen formation, and fibrosis area; and the consequent structural damage and functional impairment.

Our study has some limitations. We used a murine model of VIDD with pretreatment of bleomycin to investigate the pathogenic pathway in the process of diaphragm fibrosis in vivo and establish the critical role of PI3K-γ signaling. However, the interaction between different cell populations participating in diaphragm fibrosis and the PI3K-γ pathway, as well as the expression of fibrogenic makers within these cell populations, should be explored. Awad et al. reported that cardiomyofibroblasts and cardiomyocytes exhibit different responses to TNF stimulation in a PI3K-γ-dependent manner, contributing to myocardial remodeling [39]. Further in vitro study using the isolation and culture of primary diaphragm fibroblasts or myocytes from our injured PI3K-γ-knockout and wild-type mice is warranted to provide insights into the underlying pathophysiology and potential therapeutic targets.

## 4. Materials and Methods

### 4.1. Ethics of the Experimental Use of the Animals

Wild-type and PI3K-γ-deficient C57BL/6 mice, weighing between 20 and 25 g, aged between 6 and 8 weeks, were obtained from Jackson Laboratories (catalog number: 024587 (PI3K-γ), Bar Harbor, ME) and the National Laboratory Animal Center (Taipei, Taiwan) [42]. Briefly, homozygous mutants (PI3K-γ^−/−^) exhibit an impaired neutrophil chemotaxis and respiratory burst in response to formyl peptide N-formyl-Met-Leu-Phe and C5a induction, as well as reduced thymocyte survival and activation of mature T lymphocytes [22]. The lower expressions of the PI3K-γ protein in PI3K-γ^−/−^ mice were confirmed by using a Western blot analysis. The study was performed in strict accordance with the recommendations in the Guide for the Care and Use of Laboratory Animals of the National Institutes of Health. The protocol was approved by the Institutional Animal Care and Use Committee of Chang Gung Memorial Hospital (permit number: 2021111002). All surgery was performed under ketamine and xylazine anesthesia, and all efforts were made to minimize suffering.

### 4.2. Experimental Groups

Animals were randomly distributed into 7 groups in each experiment: group 1, nonventilated control wild-type mice with normal saline; group 2, nonventilated control wild-type mice with bleomycin; group 3, V_T_ 6 mL/kg wild-type mice with bleomycin; group 4, V_T_ 10 mL/kg wild-type mice with normal saline; group 5, V_T_ 10 mL/kg wild-type mice with bleomycin; group 6, V_T_ 10 mL/kg Pi3K-γ^−/−^ mice with bleomycin; group 7, V_T_ 10 mL/kg wild-type mice after AS605240 (5 mg/kg) administration with bleomycin. The V_T_ 6 mL/kg wild-type mice with bleomycin group was designed as the group in which the low-tidal-volume protective ventilatory strategy would be applied for comparison of the experimental parameters of the nonventilated wild-type mice with those of the bleomycin group and of the V_T_ 10 mL/kg wild-type mice with those of the bleomycin group. In each group, three mice underwent TEM and micro-CT, and five mice underwent measurements for immunohistochemistry and immunofluorescence assays, inflammatory cytokines, oxidative and antioxidative loads, Masson’s trichrome staining, and Western blots.

### 4.3. Statistical Evaluation

The Western blots were quantitated using a National Institutes of Health (NIH) image analyzer, Image J 1.27z (National Institutes of Health, Bethesda, MD), and the results were presented in arbitrary units. Values were expressed as the means ± SDs from at least 5 separate experiments. The data on MDA, histopathologic assays, and oxygenation were analyzed using Statview 5.0 (Abascus Concepts Inc., Cary, NC, USA; SAS Institute, Inc., Cary, NC, USA). All results of the real-time polymerase chain reactions and the Western blots were normalized to the nonventilated control wild-type mice with room air. ANOVA was used to assess the statistical significance of the differences, followed by multiple comparisons with Scheffe′s test, and a *p* value < 0.05 was considered statistically significant. 

Bleomycin administration, collagen assays, measurement of the cross-sectional areas of the muscle fibers, measurement of diaphragm excursion and thickness, preparation of the experimental animals, immunoblot analyses, immunofluorescence labelling, immunohistochemistry assays, measurement of inflammatory cytokines, Masson’s trichrome staining and fibrosis area measurement, measurement of MDA, administration of pharmacological inhibitors, real-time polymerase chain reactions, TEM, TUNEL assays, and the ventilator protocol were performed as previously described and detailed in the Appendix A [1,2,8,43,44].

## 5. Conclusions

Overall, we successfully induced diaphragm fibrogenesis with muscle loss in our animal model to simulate diaphragm injury in critically ill patients receiving MV following ALI. Our results indicate that the PI3K-γ pathway partially mediates the pathogenic mechanisms involved in diaphragm fibrogenesis and dysfunction. Antifibrotic therapy targeting the inhibition of PI3K-γ signaling may help to preserve the structural integrity and performance of the diaphragm as well as mitigate the consequences of diaphragm injury, such as fibrosis and muscle atrophy; however, such therapy may also reduce VIDD-associated complications and disabilities in ICU patients with ALI receiving MV.

## Figures and Tables

**Figure 1 ijms-25-06370-f001:**
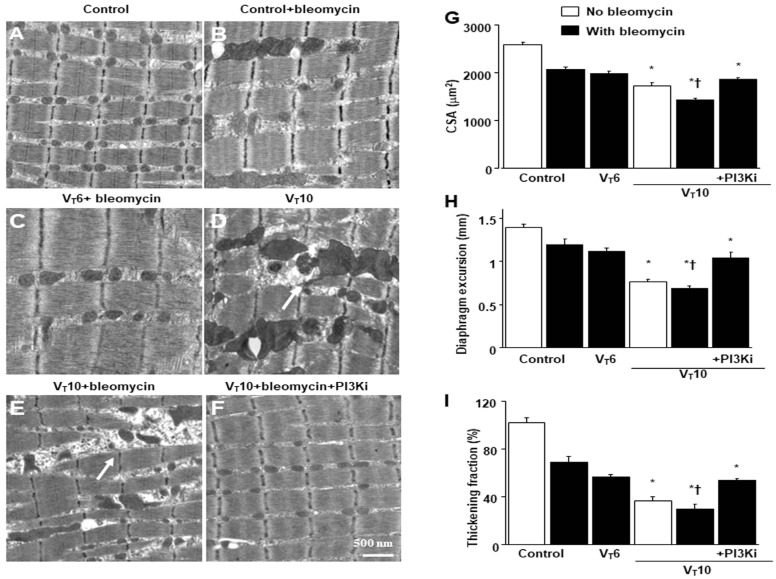
Electron microscopy, excursion, and thickening of the diaphragm. (**A**–**F**) Representative micrographs of longitudinal sections of diaphragms five days after bleomycin administration were obtained from nonventilated control mice and mice ventilated at a tidal volume (V_T_) of 6 mL/kg (V_T_ 6) or 10 mL/kg (V_T_ 10) for 8 h with room air (×40,000, n = 3 per group): (**A**,**B**) nonventilated control wild-type mice with or without bleomycin treatment: normal sarcomeres with distinct A bands, I bands, and Z bands; (**C**) 6 mL/kg wild-type mice with bleomycin treatment: reduction in diaphragmatic disruption compared to that of 10 mL/kg groups; (**D**) 10 mL/kg wild-type mice without bleomycin treatment (normal saline): increase in diaphragmatic disarray; Mitochondrial swelling with concurrent loss of cristae is identified by arrows. (**E**) 10 mL/kg wild-type mice with bleomycin treatment: disruption of sarcomeric structure with loss of streaming of Z bands, mitochondrial swelling, and accumulation of lipid droplets (asterisks); Mitochondrial swelling with concurrent loss of cristae is identified by arrows. (**F**) 10 mL/kg wild-type mice pretreated with AS605240: attenuation of diaphragmatic disruption. (**G**) Cross-sectional area of diaphragm muscle fiber was measured as described in the Methods section (n = 5 per group). (**H**,**I**) Excursion and thickness variation of diaphragm. AS605240 5 mg/kg was given intraperitoneally 1 h before mechanical ventilation. Mitochondrial swelling with concurrent loss of cristae is identified by arrows. * *p* < 0.05 versus the nonventilated control mice with bleomycin pretreatment; † *p* < 0.05 versus all other groups. Scale bar represents 500 nm. CSA = cross-sectional area; PI3KI = phosphoinositide 3-kinase-γ inhibitor (AS605240).

**Figure 2 ijms-25-06370-f002:**
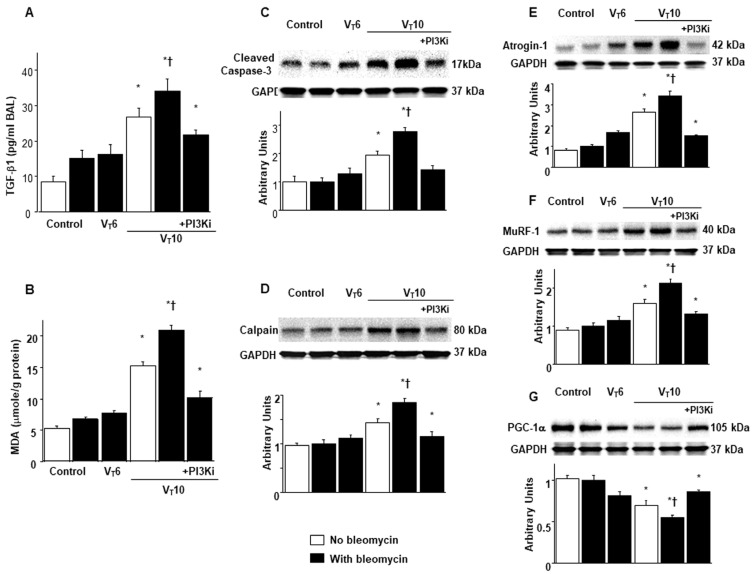
Inhibition of mechanical ventilation-mediated TGF-β1 production; oxidative stress; and caspase-3, calpain, atrogin-1, MuRF-1, and PGC-1α expression by AS605240. Five days after administering bleomycin, (**A**) BAL fluid TGF-β1 and (**B**) MDA (diaphragm) levels were observed in nonventilated control mice and those subjected to a tidal volume at 6 mL/kg or at 10 mL/kg for 8 h with room air (n = 5 per group). Western blots were performed using antibodies that recognize (**C**) caspase-3, (**D**) calpain, (**E**) atrogin-1, (**F**) MuRF-1, (**G**) PGC-1α, and GAPDH expression in the diaphragms of nonventilated control mice and mice ventilated at a tidal volume of 6 mL/kg or 10 mL/kg for 8 h with or without bleomycin administration (n = 5 per group). Relative activation was expressed in arbitrary units (n = 5 per group). AS605240 5 mg/kg was given intraperitoneally 1 h before mechanical ventilation. * *p* < 0.05 versus the nonventilated control mice with bleomycin pretreatment; † *p* < 0.05 versus all other groups. BAL = bronchoalveolar lavage; GAPDH = glyceraldehyde-3-phosphate dehydrogenase; MDA = malondialdehyde; MuRF-1 = muscle ring finger-1; PGC-1α = peroxisome proliferator-activated receptor-γ coactivator; TGF-β1 = transforming growth factor-β1.

**Figure 3 ijms-25-06370-f003:**
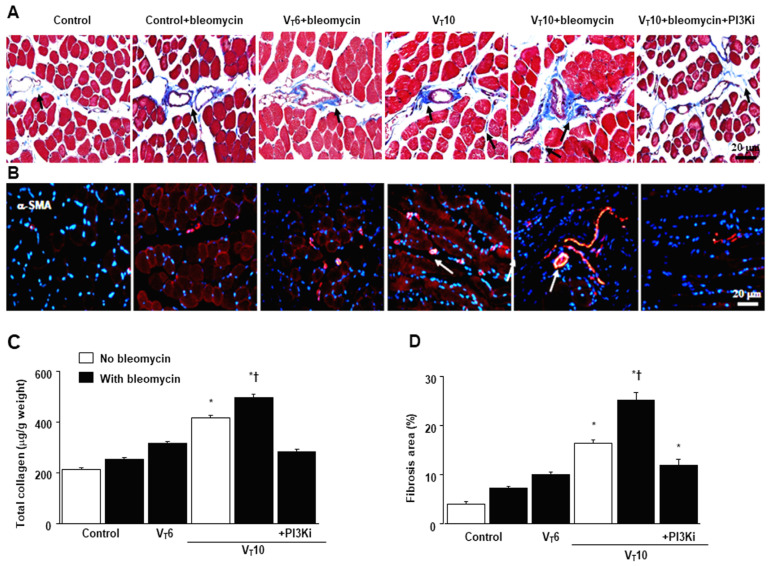
Suppression of mechanical ventilation-induced collagen production and fibrogenic markers by AS605240. (**A**) Representative micrographs (×400) with Masson’s trichrome staining. Positive blue staining of collagen in the diaphragm muscle fibers is identified by arrows (n = 5 per group). (**B**) Representative photomicrographs (×400) with α-smooth muscle actin (α-SMA, red) and Hoechst (blue) immunofluorescent staining of paraffin sections of diaphragms. (**C**) Collagen of diaphragms five days after bleomycin administration was obtained from nonventilated control mice and mice ventilated at a tidal volume of 6 mL/kg or 10 mL/kg for 8 h with room air (n = 5 per group). Positive red staining of the diaphragm muscle fibers is identified by arrows (n = 5 per group). (**D**) The fibrosis area was quantified as the average number of 10 nonoverlapping fields with Masson’s trichrome staining of paraffin diaphragm sections (n = 5 per group). * *p* < 0.05 versus the nonventilated control mice with bleomycin pretreatment; † *p* < 0.05 versus all other groups. Scale bars represent 20 μm.

**Figure 4 ijms-25-06370-f004:**
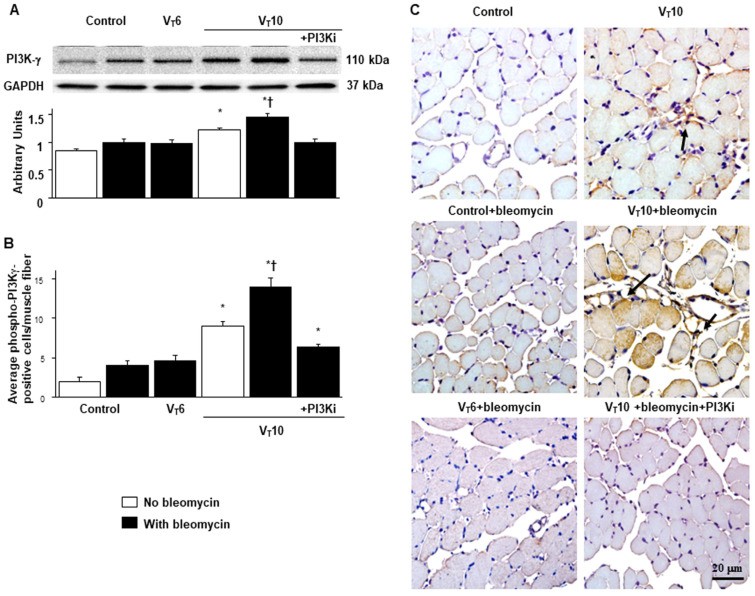
Inhibition of mechanical ventilation-induced PI3K-γ protein expression by AS605240. (**A**) Western blots were performed using antibodies that recognize PI3K-γ and GAPDH expression 5 days after administering bleomycin in the diaphragms of nonventilated control mice and mice ventilated at a tidal volume of 6 mL/kg or 10 mL/kg for 8 h with room air (n = 5 per group). Relative PI3K-γ expression was expressed in arbitrary units (n = 5 per group). (**B**,**C**) Representative micrographs (×400) with PI3K-γ staining of paraffin diaphragm sections and quantification 5 days after administering bleomycin were obtained from nonventilated control mice and mice ventilated at a tidal volume of 6 mL/kg or 10 mL/kg for 8 h with room air (n = 5 per group). AS605240 5 mg/kg was given intraperitoneally 1 h before ventilation. A dark-brown diaminobenzidine signal identified by arrows indicated positive staining for PI3K-γ in the diaphragm. * *p* < 0.05 versus the nonventilated control mice with bleomycin pretreatment; † *p* < 0.05 versus all other groups. Scale bars represent 20 μm. PI3K-γ = phosphoinositide 3-kinase-γ. A dark-brown diaminobenzidine signal identified by arrows indicated positive staining for PI3K-γ in the dia-phragm.

**Figure 5 ijms-25-06370-f005:**
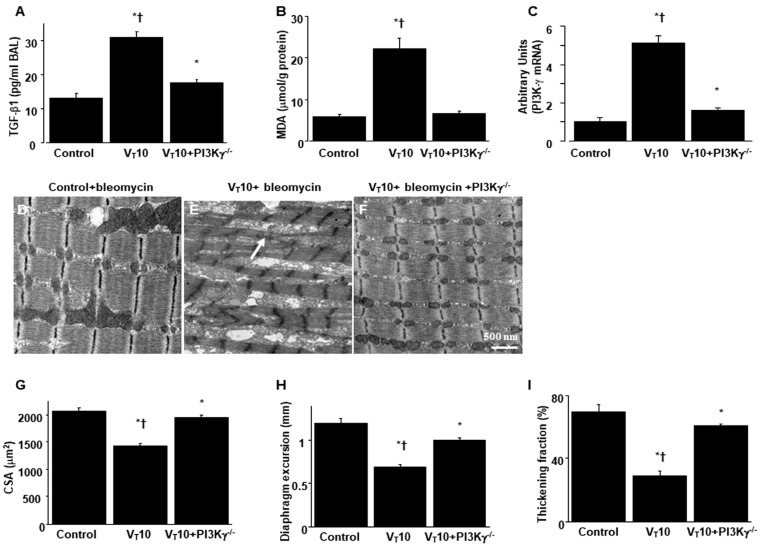
Reduction in mechanical ventilation-induced diaphragm dysfunction in PI3K-γ-deficient mice. (**A**) Results for BAL fluid TGF-β1, (**B**) MDA (diaphragm), and (**C**) real-time PCR performed for PI3K-γ mRNA expression five days after bleomycin administration were obtained from the nonventilated control mice and those subjected to a tidal volume 10 mL/kg for 8 h with room air (n = 5 per group). (**D**–**F**) Representative micrographs of the longitudinal sections of diaphragms after five days of bleomycin administration were obtained from nonventilated control mice and mice ventilated at a tidal volume of 10 mL/kg for 8 h with room air (×40,000, n = 3 per group); Mitochondrial swelling with concurrent loss of cristae is identified by arrows. (**G**) Cross-sectional area of diaphragm muscle fiber was measured as described in the Methods section (n = 5 per group). (**H**,**I**) Excursion and thickness variation of diaphragm. Mitochondrial swelling with concurrent loss of cristae is identified by arrows. * *p* < 0.05 versus the nonventilated control mice with bleomycin; † *p* < 0.05 versus PI3K-γ-deficient mice. Scale bar represents 500 nm.

**Figure 6 ijms-25-06370-f006:**
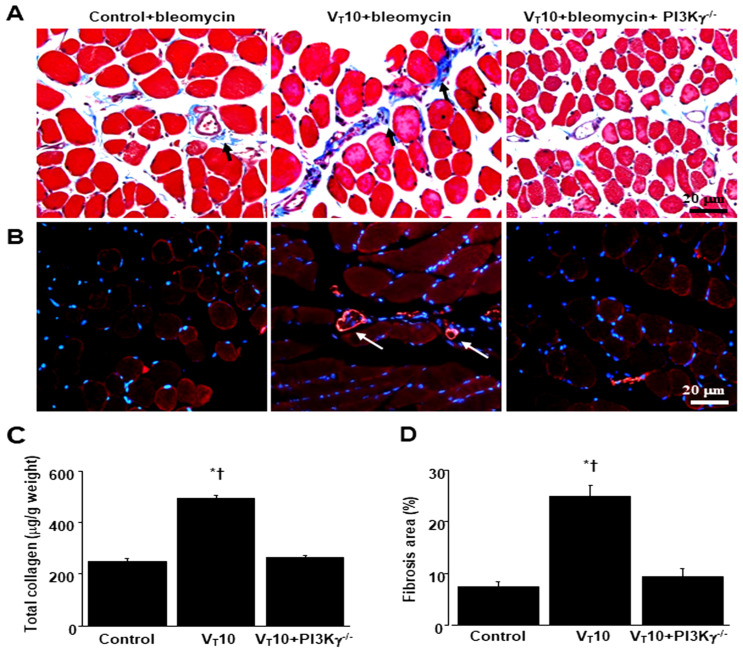
Inhibition of mechanical ventilation-induced collagen accumulation and fibrogenic markers in PI3K-γ-deficient mice. (**A**) Representative micrographs (×400) with Masson’s trichrome staining, (**B**) representative photomicrographs (×400) with α-smooth muscle actin (α-SMA, red) and Hoechst (blue) immunofluorescent staining of paraffin sections of diaphragms, and (**C**) collagen of diaphragms five days after bleomycin administration were obtained from nonventilated control mice and mice ventilated at a tidal volume of 10 mL/kg for 8 h with room air (n = 5 per group). Positive red staining in the diaphragm muscle fibers is identified by arrows (n = 5 per group). (**D**) The fibrosis area was quantified as the average number of 10 nonoverlapping fields with Masson’s trichrome staining of paraffin diaphragm sections (n = 5 per group). * *p* < 0.05 versus the nonventilated control mice with bleomycin pretreatment; † *p* < 0.05 versus PI3K-γ-deficient mice. Scale bars represent 20 μm.

**Figure 7 ijms-25-06370-f007:**
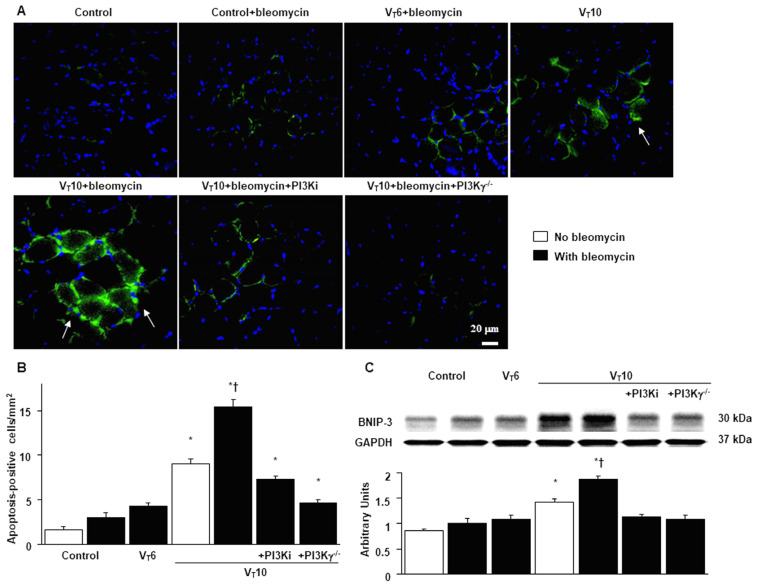
Suppression of mechanical ventilation-augmented expression of BNIP-3 and muscle fiber apoptosis by AS605240 in PI3K-γ-deficient mice. (**A**,**B**) Representative micrographs (×400) with TUNEL staining of paraffin diaphragm sections and quantification five days after bleomycin administration were obtained from the diaphragms of nonventilated control mice and mice ventilated at a tidal volume of 6 mL/kg or 10 mL/kg for 8 h with room air (n = 5 per group). A bright green signal indicates positive staining of apoptotic cells. (**C**) Western blots were performed using antibodies that recognize BNIP-3 and GAPDH expression 5 days after administering bleomycin from the diaphragms of nonventilated control mice and mice ventilated at a tidal volume of 6 mL/kg or 10 mL/kg for 8 h with room air (n = 5 per group). Relative BNIP-3 expression was expressed in arbitrary units (n = 5 per group). AS605240 5 mg/kg was given intraperitoneally 1 h before ventilation. Apoptotic cells are identified by arrows. A bright-green signal indicates positive staining of apoptotic cells, and shades of dull green signify nonreactive cells. * *p* < 0.05 versus the nonventilated control mice with bleomycin pretreatment; † *p* < 0.05 versus all other groups. Scale bars represent 20 µm. BNIP3 = BCL2/adenovirus E1B 19 kDa protein-interacting protein 3; TUNEL = terminal deoxynucleotidyl transferase-mediated dUTP-biotin nick end labeling.

**Figure 8 ijms-25-06370-f008:**
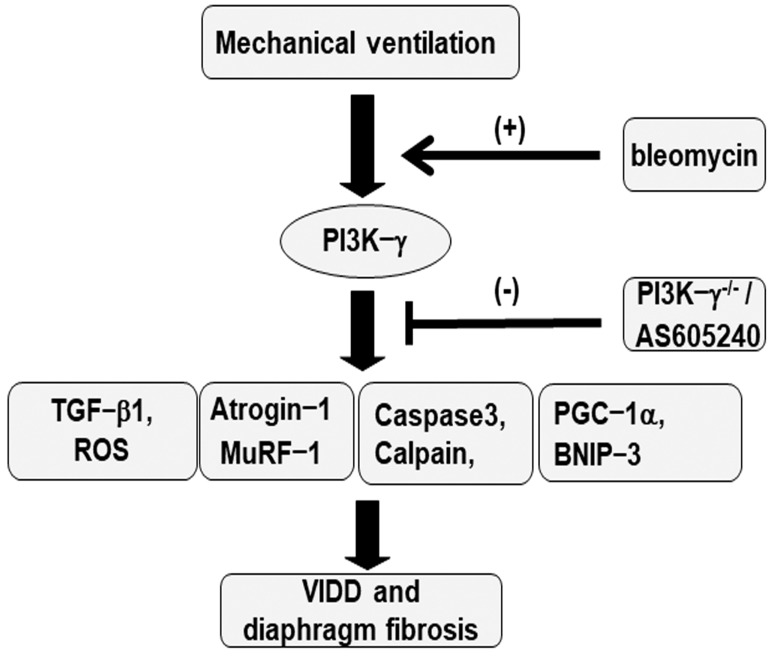
Schematic figure illustrating the signaling pathway activation with mechanical ventilation and bleomycin. Bleomycin-induced augmentation of mechanical stretch−mediated cytokine production, diaphragm damage, and fibrosis were attenuated in PI3K−γ−deficient mice and via pharmacological inhibition with AS605240. BNIP3 = BCL2/adenovirus E1B 19 kDa protein-interacting protein 3; MDA = malondialdehyde; MuRF−1 = muscle ring finger−1; PGC−1α = peroxisome proliferator-activated receptor-γ coactivator; PI3K−γ^−/−^ = PI3K−γ−deficient mice; ROS = reactive oxygen species; TGF−β1 = transforming growth factor−β1; VIDD = ventilator−induced diaphragm dysfunction.

## Data Availability

The data presented in this study are available on request from the corresponding author.

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
