# Peer review of "Suppression of Ventilation-Induced Diaphragm Fibrosis through the Phosphoinositide 3-Kinase-γ in a Murine Bleomycin-Induced Acute Lung Injury Model"

_ijms, 2024, doi:10.3390/ijms25126370_

Round 1
Reviewer 1 Report
Comments and Suggestions for Authors
This article explores how mechanical ventilation affects the diaphragm in patients with acute lung injury. The authors found that ventilation can lead to diaphragm muscle problems, and the authors discovered that Phosphoinositide 3-kinase- γ (PI3K- γ) pathway is involved in this process. This study suggests that targeting PI3K-γ could help prevent diaphragm issues caused by mechanical ventilation. This work is important for understanding and potentially treating complications of ventilation.
Major point:
The diaphragm consists of various types of cells, including myocytes (muscle cells), endothelial cells, fibroblasts, nerve cells, and macrophages and more. Among these, which cell type primarily contributes to ventilation-induced diaphragm fibrosis? Identifying the cellular players responsible for fibrotic changes within the diaphragm can provide insights into the underlying pathophysiology and potential therapeutic targets.
Conduct a thorough literature review to identify studies that have investigated the cellular mechanisms underlying diaphragm fibrosis. Look for research that examines the role of specific cell types, such as fibroblasts, myocytes, or immune cells, in promoting fibrotic changes within the diaphragm.
The authors have performed Masson's trichrome staining. It is useful to visualize and highlight the location of collagen deposition and fibrotic areas within the diaphragm.
Additionally, immunohistochemistry or immunofluorescence staining can help identify specific cell types involved in the fibrotic process. From PI3K-γ staining of paraffin diaphragm sections, which cell type predominantly exhibits PI3K-γ expression?
If possible, isolate and characterize different cell populations from diaphragm tissue samples. Analyze the expression of fibrotic markers within these cell populations to determine their contribution to diaphragm fibrosis.
Reviewer 2 Report
Comments and Suggestions for Authors
The authors have conducted a comprehensive study of ventilator induced injury with respect to diaphragm fibrosis. It is a well conducted study and here are a few suggestions to improve the manuscript.
Results:
1. All figures need to have uniform labelling. The three columns need to be label clearly so as not to confuse the reader.
2. The Vt10 line for all columns are extending to three columns including the control and the +AS605240 label covers 2 columns.
3. It would be better to have distinct patterns for control, Vt6, Vt10 and vt10+AS605240 columns throughout the paper.
4. Please describe the controls. there is control, control with bleomycin, vt6 vt10 control, vt10 with bleo and vt10+AS605240. What is the role of vt6? if there is no vt6 control what is vt6 bleo compared to?
5. figure 5 c mentions qPCR however, the methods section lacks a qPCr analysis description. Additionally, why is the qPCR data expressed in arbitrary values instead of fold change?
6. Please check the spelling of bleomycin in all figures.
7. in sma images there is a lot of background staining and needs to be checked.
7. The discussion section is quite long and can be reduced. Please condense the first paragraph which is in the form of points into a paragraph.
Round 2
Reviewer 1 Report
Comments and Suggestions for Authors
No more comments.